# Recent Advances in Stem Cell Therapies to Address Neuroinflammation, Stem Cell Survival, and the Need for Rehabilitative Therapies to Treat Traumatic Brain Injuries

**DOI:** 10.3390/ijms22041978

**Published:** 2021-02-17

**Authors:** George R. Bjorklund, Trent R. Anderson, Sarah E. Stabenfeldt

**Affiliations:** 1School of Biological and Health Systems Engineering, Ira A, Fulton Schools of Engineering, Arizona State University, Tempe, AZ 85281, USA; gbjorklu@asu.edu; 2Basic Medical Sciences, College of Medicine–Phoenix, University of Arizona, Phoenix, AZ 85004, USA; andersot@email.arizona.edu

**Keywords:** traumatic brain injury, stem cell, biomaterials, neuroinflammation, rehabilitation

## Abstract

Traumatic brain injuries (TBIs) are a significant health problem both in the United States and worldwide with over 27 million cases being reported globally every year. TBIs can vary significantly from a mild TBI with short-term symptoms to a moderate or severe TBI that can result in long-term or life-long detrimental effects. In the case of a moderate to severe TBI, the primary injury causes immediate damage to structural tissue and cellular components. This may be followed by secondary injuries that can be the cause of chronic and debilitating neurodegenerative effects. At present, there are no standard treatments that effectively target the primary or secondary TBI injuries themselves. Current treatment strategies often focus on addressing post-injury symptoms, including the trauma itself as well as the development of cognitive, behavioral, and psychiatric impairment. Additional therapies such as pharmacological, stem cell, and rehabilitative have in some cases shown little to no improvement on their own, but when applied in combination have given encouraging results. In this review, we will abridge and discuss some of the most recent research advances in stem cell therapies, advanced engineered biomaterials used to support stem transplantation, and the role of rehabilitative therapies in TBI treatment. These research examples are intended to form a multi-tiered perspective for stem-cell therapies used to treat TBIs; stem cells and stem cell products to mitigate neuroinflammation and provide neuroprotective effects, biomaterials to support the survival, migration, and integration of transplanted stem cells, and finally rehabilitative therapies to support stem cell integration and compensatory and restorative plasticity.

## 1. Introduction

In 2013 in the United States alone, approximately 2.8 million emergency department visits and hospitalizations were reported due to traumatic brain injuries (TBIs) [1]. The primary injury of a TBI results from a blow or jolt to the head or a penetrating head wound that causes functional and cognitive disabilities and an immediate disruption to the normal functions of the brain. Directly following a primary TBI injury, long-term secondary events within the cells and circuitry of the central nervous system (CNS) begin. These events include cerebral, metabolic, and mitochondrial dysfunction, oxidative stress, apoptosis, excitotoxicity, DNA damage and necrosis, and disruptions in cellular signaling [2,3]. Secondary TBI injuries have been described as a chronic disease process that includes several short and long-term detrimental effects that can dramatically increase the risk of developing neurodegenerative disorders (NDDs). The Centers for Disease Control and Prevention (CDC) reported in 1999, the last reliable date this could be found, that approximately 5.3 million individuals were living with a permanent disability due to a TBI [4]. Currently, little can be done to reverse the initial damage of a brain injury leaving medical personnel to focus on preventing further injury and providing symptomatic treatments. Long term post TBI treatments largely consist of rehabilitative therapies, both cognitive and physical, which arguably may do little to provide neuroprotective or neurorestorative benefits on their own.

Immediately following a TBI, responses from both the peripheral immune system and the neuroimmune system react to create an inflammatory response designed to promote mobilization of additional immune cells and to clear cellular debris. The body’s immune response resulting from both the initial trauma and the disruption of the blood-brain barrier (BBB) results in the infiltration of peripheral leukocytes into the injury area. Destroyed and damaged cells within the injury area release damage-associated molecular pattern molecules (DAMPS) that stimulate the release of pro- and anti-inflammatory cytokines and reactive oxygen and nitrogen species. The DAMPS are recognized by microglia, astrocytes, neurons and the infiltrating peripheral immune cells which respond by creating the neuroinflammatory response [5,6]. Both the peripheral and neuroimmune responses create a feedback loop that further releases pro- and anti-inflammatory molecular mediators, metalloproteinases, and oxidative metabolites that cause additional damage and perpetuates the neurodegenerative state following the TBI. Together, these immune responses can prolong and exacerbate the pathological response to the injury furthering the risk of developing NDDs [5,7].

To combat the initial and ongoing immune responses following a TBI, research within the field of TBIs has focused on alleviating the neuroinflammatory state using pharmaceuticals and stem cell therapies both separately and combined. In recent years, the idea of combinatorial therapies using stem cells and pharmacological agents to treat TBIs has shown a great deal of promise in managing the inflammation processes resulting from a TBI. A recent review article by Bonsack et al., calls attention to several preclinical studies using a combination of pharmacological and stem cell transplantation strategies to treat TBIs [8]. That review concluded that within those referenced animal studies, the combination of pharmacotherapy and stem cell replacement therapies was able to provide the best outcomes compared to each of these individual therapies on their own. The combination of pharmacological and stem cell therapies was effective in reducing neuroinflammation while the stem cell therapy had the potential abilities of replacing lost brain cells, secreting neurotrophic factors, and recruiting cytokines and endogenous stem cells to the injury area. Bonsack concluded that prior to stem cell combination therapies being translated to the clinic, further research is warranted to ensure safety in larger animals models, the optimization of treatment parameters and the design of more effective controlled animal studies [8].

While these strategies have shown efficacy in mitigating the intense and prolonged effects of the inflammatory response, the regenerative effects of stem cell therapies are not being fully realized due to low survival and integration rates. This issue is being addressed by researchers through the use of advanced engineered biomaterials designed to mimic the native extracellular matrix and provide a scaffold to support the nascent stem cell transplants. The incorporation of these materials to aid the transplanted stem cells is shown to support the survival, migration, and integration of the stem cells within the native neural tissue. To further enhance the regenerative capabilities of transplanted stem cells and enhance integration and neural plasticity, physical and cognitive rehabilitative therapies are employed soon after transplantation. These three treatment processes: control or mitigation of the neuroinflammatory response post TBI through the use of stem cells and/or the products of stem cells; ensuring the survival and fitness of engrafted stem cells through bioengineered extracellular environments; and providing the stimulus needed for integration and neural plasticity by cognitive and physical rehabilitation, can possibly be considered as a crucial therapy structure in the near and long term treatment of TBIs.

Recently published reviews have documented the use and efficacy of stem cell replacement therapies, pharmacological treatments, and combination therapies in the preclinical and basic research settings to treat TBIs [8,9,10,11,12,13]. This current review, however, is intended as a perspective focusing on the three therapeutic points mentioned through selected and recent preclinical and basic research in the efficacy of combinatorial stem cell therapies used to treat TBIs. First, we will highlight and recap current unreviewed studies in stem cell efficacy treating the post TBI inflammatory response. Next, we will present the latest studies involving the use of advanced biomaterials to support stem cell transplantation, retention, and integration. Finally, we will address the need for physical and cognitive rehabilitation to support stem cell integration and neural plasticity. This hierarchical combinatorial treatment approach will serve to address the immediate need for neuroprotection, a necessity for cell replacement, and circuit restoration following a TBI.

## 2. Neuroprotection—Stem Cell Therapies Targeting Neuroinflammation Following a TBI

Stem cell-based therapies to treat neuroinflammation in both the acute and chronic phase of TBIs is an approach that has the potential to mitigate secondary cell loss and to promote neurological and functional recovery. Several very recent reviews document the testing, use, and efficacy of pharmacological and stem cell replacement therapies both separately and in combination [8,9,10,11,12,13,14,15,16]. Here, our review serves to highlight some of the most recent research reports in the use of stem cell therapies alone to treat the destructive and chronic neuroinflammatory processes of TBIs. The following most recent research articles further demonstrate several crucial points in the application of stem cell therapies to treat TBIs including the choice of stem cell type (or stem cell products), the temporal aspects of stem cell therapies, and the delivery route involved.

### 2.1. Recent Stem Cell Studies Involving Mesenchymal Stem Cells (MSCs)

Mesenchymal stem cells (MSCs) have displayed a robust ability to modulate the inflammatory response within the immediate area of a TBI and have received a great amount of attention in research in the last several years. MSCs are multipotent fibroblast-like cells that can be easily isolated from a multitude of adult tissues, avoid the controversial and ethical issues surrounding the use of human embryonic stem cells (hESCs), and most importantly display a wide range of immuno-modulatory capabilities [17,18,19,20]. Currently however there is some controversy surrounding safety issues regarding MSCs. The same immunosuppressive effects displayed by MSCs may lead to the possibility of inadvertent tumor growth or metastasis by the sweeping suppression of the immune and antitumor response. Further safety concerns include several factors such as the source, handling, and culturing of cells, heterogeneity of cells, route of administration, i.e., intravenous versus direct injection and number of cells and concentrations [21,22,23,24,25]. In addition to the need for more preclinical research including longer term studies, currently there are several ongoing clinical studies with the objective to assess the safety and efficacy of the use of MSCs in the treatment of TBIs [26].

#### 2.1.1. Intravenous Application of MSCs, MSC Exosomes, and MSC Secretomes Following a TBI

Several previous studies have shown the effectiveness of using MSCs to treat TBIs by characterizing the cell source, delivery route, dosage, and timing of delivery. However, there is still little known concerning the effects of early verses delayed treatments using MSCs. Using human adipose-derived mesenchymal stem cells (HB-adMSCs), Ruppert et al. (2020) explored the neuromodulatory potential of early and delayed administration as a treatment for TBIs [5]. The delayed treatment design of Ruppert’s study is meant to resemble the time it would take to isolate, expand, characterize, and deliver cells in the absence of previously banked cells. Rats were subjected to a moderate to severe TBI using a controlled cortical impact (CCI) followed by the intravenous infusion of HB-adMSCs at 3- and 14-days post injury (DPI). Their results indicated that both the early and delayed HB-adMSC infusion groups benefitted significantly. Early HB-adMSC administration at 3 DPI effectively diminished M1 microglia and may have promoted neurogenesis in the hippocampal subgranular zone as indicated by doublecortin (DCX) staining and comparison with controls. Delayed HB-adMSC administration at 14 DPI indicated an anti-inflammatory drift as evidenced by a significant increase in the percentage of microglia displaying M2 markers accompanied by a significant increase in the M2/M1 microglia ratio. Additionally, they found that HB-adMSC administration at both early and delayed time points significantly improved spatial memory performance in the Morris water maze. The data from this study indicated that the therapeutic benefits of performing MSC stem cell infusions for the treatment of TBIs generated significant beneficial results over a relatively wide window of treatment times.

In addition to intravenous infusion of MSCs to modulate the inflammatory response to TBIs, several recent studies have explored the intravenous application of the MSC exosome and secretome [27,28,29,30]. Unlike the intravenous injection of whole MSCs, the exosomes are able to pass the BBB to convey their contents to recipient cells without the risk of vascular obstruction or risks of tumorgenicity [27]. In this recent study, Zhang et al. (2020) set out to determine the dose and time dependent efficacy of MSC derived exosomes following a TBI [27]. Zhang had previously shown that MSC derived exosomes could significantly improve functional outcomes following a TBI [31]. To test dose-responses, exosomes were collected from the supernatant of cultured MSCs and injected intravenously in rats at 50, 100, and 200 µg doses at 1 DPI. To test a therapeutic window, 100µg doses of exosomes were injected at 1, 4, and 7 DPI [27,32]. Results of these studies showed that the therapeutic effects of the 100µg dosage were significantly greater than the 50 or 200µg doses and that functional and histological outcomes were significantly greater when 100µg of exosomes were administered at 1 DPI verses 4 or 7 DPI. Overall, this study showed that regardless of the dose of exosomes used or the delay in treatment, there were significant improvements in cognitive and sensorimotor function, reduced neuroinflammation and hippocampal neuronal cell loss, and an increase in neurogenesis and angiogenesis [27].

To further probe the therapeutic effects of MSC-derived exosomes, Xian et al., (2019) employed both in vivo and in vitro models to determine the effects on neuroinflammatory processes, aberrant calcium signaling, and mitochondrial dysfunction [29]. Cultured hippocampal astrocytes were stimulated with lipopolysaccharide (LPS) to induce an inflammatory response, then treated with prepared MSC exosomes. In this experiment, it was found that application of the MSC exosomes significantly reduced the upregulation of the proinflammatory cytokines tumor necrosis factor-alpha (TNF-α), interleukin-1 beta (IL-1β), and reduced levels of CD81, a regulator of astrocytic activation. It was further found that MSC exosome treatment reversed altered calcium signaling by decreasing Ca^2+^ influx and ameliorated mitochondrial dysfunction in the LPS-induced astrocytes. While not a model of traumatic brain injury, the in vivo experiments utilized a pilocarpine induced mouse model of status epilepticus (SE) to examine the therapeutic effects of MSC exosomes on hippocampal inflammation. The MSC exosome treatment here also significantly reduced TNF-α and IL-1β in the SE-induced group when compared to the control group. Furthermore, the MSC exosome treatment significantly reduced TNF-α and IL-1β RNA levels in the SE-induced group compared to the control group [29].

A separate group, Baez-Jurado et al. (2018) evaluated cultured astrocytes that were subjected to a scratch injury and subsequently treated with conditioned culture medium from human adipose-derived mesenchymal stem cells (CM-hMSCA). Interestingly, Baez-Jurado found similar result to those reported by Xian in that intracellular Ca^2+^ levels were reduced and evidence of maintained mitochondrial dynamics [30]. Baez-Jurado further found that treatment with CM-hMSCA reduced the levels of proinflammatory cytokines IL-6, TNF-α, and granulocyte-macrophage colony-stimulating factor (GM-CSF) and elevated levels of neuroprotective cytokines IL-2 and IL-8 [30].

While the previous studies used purified exosomes of the cultured MSCs, a study by Xu et al., (2020) asked whether the intravenous application of the entire secretome of cultured MSCs would improve neuroinflammation and functional outcomes following a TBI [28]. For this study, the culture media of adipose-derived MSCs was desalted and concentrated prior to intravenous injection into rats. Injections were performed starting at 1 DPI and performed daily for 7 days following a TBI. They found that the MSC secretome injections were effective in mediating post TBI neuroinflammation by promoting microglial switching from M1 to M2 phenotypes, lowering the secretion of inflammatory cytokines, mitigating neural cell apoptosis, and relieving brain tissue edema [28].

#### 2.1.2. Direct Transplantation of MSCs Following a TBI

The transplantation of MSC cells overexpressing cytokines is an emerging approach to control neuroinflammation and promote a change from destructive inflammatory processes to restorative. Enam et al. transfected MSCs to transiently induce the expression of IL-4 and thereby induce macrophages and microglia to an M2 state and prompt astrocytes to secrete growth factors when transplanted following a TBI. In vitro experiments showed that the IL-4 expressing MSCs could induce an M2 phenotype in macrophages as shown through the detection of the CD206 cell surface marker. However, as Enam concluded with their in vivo experiments, while this method did induce cytokine and gene level changes, it was not sufficient to change any functional or histological outcomes following a TBI. Additional evidence from transcriptomic studies revealed persistent inflammatory pathway conditions and a lack of neuroregeneration [33].

Additional recent studies involving the direct transplant of genetically modified MSCs to the area of a TBI, were performed by Peruzzaro et al., (2019) and Maiti et al., (2019) [32,33]. Each of these studies used bone marrow-derived MSCs that were virally transfected to overexpress human IL-10, an anti-inflammatory cytokine. The Peruzzaro study was performed to determine if TBI+MSC or TBI+MSC+IL-10 would reduce inflammation, produce pro-immunomodulatory effects, or improve functional outcomes following a TBI. Their results indicated a significant increase in the levels of the proinflammatory cytokine TNF-α in the TBI+MSC group when compared to the sham+vehicle and TBI+MSC+IL-10 groups. The authors attribute this result to a possibly more pronounced inflammatory response in the TBI+MSC group but do not discuss the possibility of the effects of IL-10 overexpression on TNF-α. Additional evidence of the actions of the transplanted MSCs or MSCs+IL-10 were shown in the reduction of the inflammatory marker CD86 compared to the TBI+vehicle group and the increase in the percentage of CD163 expressing cells indicating a shift to M2 phenotypes. Functional outcomes indicated by testing using the Morris water maze, a ladder rung walking task, and rotarod did not provide any significant improvements when the TBI+MSC or TBI+MSC+IL-10 groups were compared to a sham plus vehicle group. Measurements of tissue sparring also failed to find any significant differences between these same groups. An interesting finding of this study is that after 3 weeks post injury, no MSCs were found in either the TBI+MSC or TBI+MSC+IL-10 groups [34]. In a separate study, Maiti et al. performed experiments similar to Peruzzaro’s but instead investigated levels of autophagy, mitophagy, molecular chaperones, neuroinflammation, cell death, and synaptic functioning following a TBI and treated with MSCs or MSCs overexpressing IL-10. Maiti reported that mild to moderate neuroprotective effects after transplantation of MSCs+IL-10 or MSCs alone following a TBI. These effects included increases in autophagy marker detection, greater mitophagic cell survival, and an increase in pre- and post-synaptic integrity. These were accompanied by a decrease in the detection of cell death markers. Although Maiti reported that the MSC-IL-10 transplants exhibited greater neuroprotective effects than MSC transplants alone, no mention of transplant survival was found [35].

### 2.2. Embryonic Cell Transplants and Tissue Engraftments

Using embryonic motor cortex tissue transplants in a mouse model of TBI, Ballout et al., (2019) sought to characterize the effects of a 1 week delay between a cortical lesion and transplantation on inflammation and the survival and development of the engrafted neurons [36]. The research group reported a significant increase of GFAP+ astrocytes, Iba1+ microglial cells/macrophages, Olig2+ oligodendrocytes, and CD45+ hematopoietic cells at 7 DPI post lesion compared to controls and day 0 groups. They further reported that a microglial/macrophage transition from M1 to M2 states at 7 DPI followed by an M1 response at DPI indicating the transient nature of the M2 microglial state. When comparing the embryonic tissue transplants performed at days 0 (no delay) and 7 (delay), no differences were found in the numbers of microglial or hematopoietic cells in the cortex or within the transplant itself. However, a significant increase of GFAP+ astrocytes were found in both the cortex and in the transplant of the delay group when compared to the no delay group. Furthermore, no differences were found in the number of Olig2+ expressing oligodendrocytes in the cortex of the groups but a significant increase was detected in the transplant of the delay group compared with the no delay group. No differences were detected in the numbers of A1 astrocytes cortex or transplant of either groups but a there was a significant increase in the percentage of A2 astrocytes in the cortex of the delay group and M1 microglia only decreased within the transplant. Taken together, the results of this study indicated a favorable advantage regarding neural tissue and functional recovery of the 1-week delay for the transplantation of the cortical tissue. According to the authors, the results further suggest that the neuroinflammation observed at 7 DPI lesioning may be promoting clearance of debris, vascularization of the grafted tissue and by the development of projections and myelination of axons of the transplanted neurons.

## 3. Regeneration—Engineered Biomaterials to Support Stem Cell Survival and Engraftment

Broad advances have been made in stem cell therapies to treat a myriad of disorders over the last few decades. However, especially in the case of TBIs, stem cells themselves still face the difficulties of survival, migration, and engraftment. The neuroinflammatory environment of a TBI is marked by reactive microglia and astrocytes and the invasion of leukocytes from the peripheral immune system creating an injury climate that is not at all conducive to those processes. Furthermore, there has been a reported correlation between high stem cell survival rates and decreased levels of pro-inflammatory cytokines as well as a shift from an M1 pro-inflammatory state towards an M2 resolution phase of inflammation [37]. To create a more permissive setting, or shield stem cell transplants from the destructive inflammatory process, research has turned to bioengineered solutions to mimic the native extracellular environment in order to support the stem cell transplants. In this context, these 3D scaffolds made from natural and synthetic polymers and peptides are designed to serve as a scaffold or platform for cell and/or bioactive molecules which are delivered to the injury site. Hlavac et al., (2019), Tan et al., (2019), and Nikolova and Chavali (2019) have all provided recent and thorough reviews on the advances in biomaterials and 3D bioactive matrices for the treatment of TBIs [38,39,40,41]. This review however will again focus on recent in vivo advances in the use of these materials for the purposes of stem cell transplants and in the case of Liaudanskaya et al., (2019), seeding the stem cell scaffold with a corticosteroid to further control neuroinflammation and improve cell survival.

### 3.1. Stem Cell Transplantation within a Peptide Nano Scaffold

Sehab Negah et al. (2019) used a prepared a 3D peptide nano-scaffold, RADA4GGSIKVAV (R-GSIK), to enhance and investigate the grafting environment when seeded with human meningioma stem-like cells (hMgSCs). When used in treating a TBI, the scaffold was tested for neuroinflammatory responses, apoptosis, gliosis, the proliferative state of transplanted cells, and functional outcomes [42]. Neurological outcomes as indicated by a modified neurological severity score (mNSS), showed that both the hMgSC only and the hMgSC+R-GSIK groups performed significantly better than the TBI, PBS, and R-GSIK controls groups. Astrogliosis and reactive microglia around the lesion site were significantly reduced in the hMgSC+R-GSIK groups when compared to the hMgSC only and other control groups. Neuroinflammation as measured by the levels of TLR-4, TNF-α, and IL-1β were significantly lower in both the hMgSCs and hMgSCs+R-GSIK groups. The survival rate of the hMgSCs+R-GSIK transplanted cells was assessed by BrdU staining and found to be significantly higher that of the hMgSCs only transplants. Furthermore, the hMgSCs+R-GSIK displayed integration with the host tissue and significantly lowered the lesion volume compared to all other groups. No significant differences were found between groups in the differentiation of the transplants as indicated by olig2 and NEUN staining [42].

In a similar following experiment, Sahab Negah et al., (2020) used MSCs and the 3D peptide nano-scaffold, RADA4GGSIKVAV (R-GSIK) following a TBI to investigate the effects on functional recovery and neuroinflammation [43]. Behavioral assessments to determine functional improvements consisted of an mNSS assessment, open field (OF), and elevated plus maze (EPM). Transplant effects on neuroinflammation was assessed by measurement of proinflammatory cytokines and glial activity. Neurological deficits as assessed by the mNSS scores showed that the MSC+R-GSIK group performed significantly better than control groups at 28 DPI. Activity levels and anxiety-like behaviors as measured in OF and EPM, also indicated significant improvements over control groups. Astrogliosis and reactive microglia in the MSC+R-GSIK group was found to be significantly reduced in the area of the injury as detected by Iba-1 and GFAP staining and compared to MSC only transplant controls. Proinflammatory cytokines, TLR-4, IL-6, and TNF-α were all found to be significantly lower in both the MSC only transplant group and the MSC+R-GSIK group than control groups. No information was provided that pertained to transplant cell survival. Overall, these reported findings indicate that MSCs and the R-GSIK scaffold improve functional recovery and help to attenuate inflammation following a TBI.

### 3.2. Stem Cell Transplantation Using a Hydrogel-Based Scaffold

To better cope with the environment within the area of the TBI and promote the therapeutic effects of stem cell transplants, Yao et al. (2019) developed a thermosensitive hydrogel based on chitosan, hydroxyethyl cellulose, hyaluronic acid, and b-glycerophosphate (CS-HEC-HA/GP) [44]. This composition according to the authors, can preserve a liquid state below 25 °C and transforms to a hydrogel at 37 °C with similar rheological characteristics as that of brain tissue. Therapeutic effects of the CS-HEC-HA/GP loaded with human umbilical cord mesenchymal stem cells (hUC-MSC) were evaluated in a moderate TBI model in Sprague Dawley rats. Transplantation of the hydrogel scaffold and hUC-MSCs was performed at 7 DPI with retention, survival, and migration attributes measured at 14- and 28-days post-transplant. Measurements of lesion volume of the MSC+ scaffold treatment was significantly decreased when compared to saline treated, scaffold only treatment and MSC only treatment. Immunofluorescence (IF) comparisons between samples from MSC only and MSC+scaffold treatments indicated that the CS-HEC-HA/GP scaffold was effective in protecting the hUC-MSCs for as long as 28 days. Additionally, increased IF detection of MSCs in the hippocampus of the Msc+scaffold group compared to the MSC only group suggested an increase in migration. Testing results in the MWM showed that both the MSC and MSC+scaffold groups significantly outperformed the saline and scaffold only groups in latency reduction in the spatial navigation trial, significantly increased time in platform quadrant in the spatial probe trial, and significantly increased numbers of platform crossing on day 28 post transplantation. These results indicated that the hUC-MSC loaded CS-HEC-HA/GP scaffold provided several beneficial results following a TBI when compared to the control treatments [44].

### 3.3. Stem Cell Transplantation Using a Hydrogel-Based Scaffold with Integrated Corticosteroid

In addition to creating an artificial extracellular environment, biomaterial scaffolds are also capable of delivering additional factors along with the cellular cargo. Instead of relying solely on the addition of stem cells to modulate neuroinflammation, Liaudanskaya et al. (2019) integrated the corticosteroid methylprednisolone (MP) into a hydrogel infused silk scaffold seeded with embryonic rat cortical neurons [37,45]. Initial studies without MP integration in the silk-hydrogel scaffold showed that there was no significant neuronal survival when transplantation was performed immediately after an injury or when transplantation was delayed by 7 days and compared to seeded scaffold constructs placed on an uninjured brain. Results when MP was integrated in the scaffold showed a significant increase in survival during the acute phase of inflammation following a TBI when compared to non-MP scaffolds. The authors attribute this high survival rate in the MP integrated scaffold from the downregulation of upstream targets for most of the inflammatory cytokine and chemokine regulators. Further observations indicated the presence of reactive astrocytes or possibly radial glia or neural stem cell recruitment and proliferation due to co-activation of Notch1, STAT3, and NF-κB within the MP transplants [37].

## 4. Rehabilitative Therapies—Stem Cell Incorporation, Neural Plasticity, and Circuit Regeneration

Apart from the ability for stem cells and stem cell factors to mitigate neuroinflammation and the challenges of transplanted stem cell survival, comes the case for circuit regeneration and repair and neural plasticity post-TBI. In addition to applying physical and cognitive rehabilitation therapies to reinforce integration and post-injury experience-dependent plasticity of the cell transplants in the native tissue, functional improvements post TBI may be considered the product of a relearning process. The way in which neural circuits adapt and recover to relearn or encode skills and behaviors during repair remain essentially the same basic neurobiological processes to initially acquire skills and behaviors. [46]. Indeed, while increased transplant survival and integration may support functional recovery after a TBI, relying on a cell-based or pharmacotherapeutic approach alone will almost certainly fall short of an optimal potential outcome. Accordingly, to produce the maximal functional recovery outcome, physical and cognitive therapy approaches must be combined with planned cell transplantation therapeutics. The following summaries of preclinical data illustrate the compensatory and restorative abilities post injury that highlights the need for rehabilitative therapies to aid in the cellular and synaptic integration of stem cell transplants.

### Post Stem Cell Transplant Rehabilitation and Neural Plasticity

Despite a noted need for compensatory and restorative plasticity in the residual neural tissue, the current research space contains very little in the way of physical and cognitive rehabilitative therapies following stem cell therapies used to treat TBIs. Adkins et al. (2015) also made note that neuroplasticity and neurorehabilitation studies focusing on TBIs were lacking. This research team set out to investigate the efficacy of the same motor rehabilitation used successfully and extensively in animal models of stroke in a TBI animal model. The rehabilitation efforts consisted of skilled forelimb reaching, voluntary exercise, and uninjured forelimb constraint individually and in combinations. Their data taken together indicated that the combination of these rehabilitation tasks provided the optimum behavioral outcome. Furthermore, even though not all animals significantly improved their successes in the reaching task, many of the animals that received rehabilitation reached with more normal patterns rather than with compensatory-like or abnormal actions. This indicated that functional movements appeared to reflect an amount of true recovery in the TBI animals that received any type of rehabilitation [47].

Using the same rehabilitative efforts as Adkins, Combs et al. (2016) set out to explore a relationship between behavioral improvements and functional cortical organization post TBI. Results indicated that rehabilitative training significantly improved behavioral deficits induced by TBI. As noted in the Adkins study, abnormal reaching strategies were significantly reduced in the training group when compared to the no training group. Results of intracortical microstimulation (ICMS) mapping of the caudal motor cortex of the injured brain hemisphere showed significantly increased total area in wrist area representation but not in elbow area representation of the training group versus the no training group. While these were the only two motor cortex area representations presented, the total number of movements reported during microstimulation for wrist/digit, elbow, shoulder, neck, jaw, or trunk were not found to be significantly different between the training groups. The authors conclude that while the brain does possess the capability of post TBI regeneration, plasticity, and reorganization, it may be limited within the peri injury area of the cortex. Furthermore, neuroplasticity (post TBI) requires an extremely focused and intense rehabilitation effort to prompt remodeling and recovery [48].

A 2017 study performed by Pruitt et al. sought to characterize cortical map plasticity in the intact hemisphere of a unilateral TBI in rats that had received forelimb training prior to and after a TBI. Previous studies cited by the authors had observed that extensive rehabilitation following a stroke prompted a transient reorganization in the intact brain hemisphere that was believed to support recovery. The authors also noted that few studies had investigated plasticity within the intact hemisphere related to recovery in animal models of TBI. By performing ICMS, the team found that the unilateral TBI had significantly reduced the representation of the impaired forelimb in the uninjured motor cortex despite extensive forelimb training. This observation suggested that the TBI had acted to prevent any map reorganization seen in uninjured animals. These results were explained to be due to TBI disruption of interhemispheric connections through the corpus callosum. This suggested to the researchers that the relevant plasticity occurs specifically in the callosal projections as descending fibers in the uninjured hemisphere remained intact [49]. Whether TBI relevant plasticity occurs within the corpus collosum itself, as the authors state, or is rather restricted by the disruption of intra and interhemispheric connectivity is not clear. However, as the results of this single study suggest, additional regenerative therapies are needed to support plasticity following a TBI.

## 5. Conclusions and Perspective

This concise review of current research regarding stem cell treatments for TBIs, has meant to highlight new and central research in the use of stem cell therapies to treat TBIs. Each of these individual preclinical treatments are presented as a possible hierarchical program in the treatment for TBIs. As presented, they include: control of the immediate post injury conditions (neuroinflammation and tissue survival) through the use of stem cell or stem cell products, engineered environments for the survival and migration of stem cell transplants in the TBI environment, and the need for supportive cognitive and physical rehabilitation therapies to support neural circuit repair and compensation and neural plasticity. Current preclinical research continues to determine the safety and efficacy of a variety of stem cell types and treatment options that are safe and effective under diverse conditions presented in TBIs.

The aftermath of a moderate to severe TBI leaves a very complex immune compromised environment that leads to additional and continuing damage beyond the initial trauma. Recent stem cell studies as outlined here have shown the ability of various types of stem cells and stem cell factors to mitigate the adverse effects of the neuroinflammation process. They have furthermore demonstrated the ability to aid in the temporal transition from a destructive inflammatory response to a restorative response. With the addition of extracellular mimicking biomaterial matrices, stem cell transplant survival, retention, and migration have been significantly increased furthering the therapeutic effects of stem cell therapies. Additionally, the biomaterial scaffolds have the added capacity to deliver neurotrophins, cytokines, and/or pharmacological agents directly to the injury area. This further makes the use of stem cell seeded biomaterials used for transplantation not only advantageous to control neuroinflammation, but also offering the opportunity for neural regeneration beyond the brains own innate abilities.

A key foundation for functional improvement following a TBI is compensatory and/or restorative plasticity within the residual neural tissue [46]. As demonstrated by Combs and Pruitt, recovery of motor function following cortical injury is accompanied by a reorganization of motor cortical areas [48,49]. Combined therapies designed to augment recovery, stem cell and rehabilitative therapies, also play a part in in the enhancement of motor map plasticity and ultimately functional recovery [50,51]. While it is evident that there is an absence of studies addressing the need of specific physical and cognitive rehabilitation following stem cell transplantation, and that there currently remains a compelling need for extensive research in this area.

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
