# Peer review of "Recent Advances in Stem Cell Therapies to Address Neuroinflammation, Stem Cell Survival, and the Need for Rehabilitative Therapies to Treat Traumatic Brain Injuries"

_ijms, 2021, doi:10.3390/ijms22041978_

Round 1

Reviewer 1 Report

This review covers a vast amount of emerging research to indicate progress in the area of stem cell therapies for TBI. Unfortunately, interpretation of the body of research presented here is not comprehensive or cohesive. Findings from the emerging research are not contextualised or compared. It is therefore difficult to derive any key take home messages or understanding of what research should be performed next, or even where the research landscape is currently at regarding both rehabilitation, cell therapies or combination interventions.

Specific recommendations and reflections

Overall the tone of the paper infers that there is a lot of clinical work and that combined cell therapy for TBI is a well researched area. However this is a huge flaw of the review. There is a lack of robust clinical evidence and there remains vast cell combination candidates still being investigated preclinically. This isn’t to say that we aren’t on the cusp of existing clinical work to uncover the usefulness of these interventions. This is a promising area and research is extremely needed. But I think some of the language needs to be tempered throughout to discuss where the research is up to, and claims on safety and efficacy must be prefaced with the context the research has been carried out (e.g. preclinical).   

  • Line 23: Additional therapies such as pharmacological, stem cell, and rehabilitative have shown little to no improvement on their own, but when applied in combination have given encouraging results.

What about the STEMTRA trials? There have also been a number of clinical trials in this space using cord blood/bone marrow MSCs that have improved outcomes, even if effect sizes are small.

  • Line 56: Several very recent reviews document the use and efficacy of stem cell replacement therapies, pharmacological treatments, and combined stem cell and pharmacological therapies being tested and used

References here cover a number of neurological conditions. This section and the claims to follow are all based on documented work from animals. There needs to be a description of where the research is up to, or at least specify that the efficacy is being investigated remains in the preclinical stage and what has been done so far in the clinical area. For example, Bonsack (ref 7) is referred to on multiple occasions however the focus of the review discusses the important preclinical investigations that have primed some clinical research areas. The paper urges that more work is to be done, and clinical work to date is limited. Your next line of “That review concluded that the combination of pharmacotherapy and stem cell replacement therapies was able to provide the best outcomes compared to each of these on their own. Both of these therapies were effective together in reducing neuroinflammation while the stem cell therapy added the additional abilities of replacing lost brain cells, secreting neurotrophic factors, and recruiting cytokines and endogenous stem cells to the injury area. [7].” Will need to be significantly tempered to contextualise the findings.

  • Text under header 2 can be condensed. It is very repetitive from the introduction and i would recommend this be combined with the section prior to it. Instead this section likely needs an introduction to the candidate therapies themselves.

  • 1- Some statements presented here are false. “Whether this concern is valid only for intravenous application or for locally transplanted MSCs is not completely clear.”

It is well known than MSCs’ potential for tumour formation is based on a range of factors, not just route but also cell source, manufacture and manipulation rather than the route of administration. The referenced material is explicit about this e.g. Hasan 2017 discusses how tumor formation is a consideration, particularly for cells that have been cultured for long periods of time.

  • My suggestion is to alter and reconsider some of your discussed rationale. The field has research on stem cell therapies from a range of sources, using a range of protocols which expand and characterise cells, mainly MSCs. This heterogeneity is a leading cause of the concerns, where you cannot standardise the risk or completely address safety concerns in the field without seeing a cell candidate through the entire pipeline in a streamlined and homogenous manner.

  • In section 2, you separate the use of cells transplanted via different routes contains no discussion of the reasoning behind the different delivery methods. Instead it is just a summary of the studies which use the methods. I would suggest discussion of the rationale for both uses of these methods, or why they may differ, particularly when discussing MSCs which as you say have a well defined secretome and known trophic effects.

This information is critical as you then lead into section 3, where it is important to understand if engraftment is even essential to see functional improvements and recovery or why this would be of benefit.

  • By the end of section 3, it is unclear what method of cell delivery or use of scaffolds or combination therapy are preferred or show better outcomes. The sections do summarise the information but there are no comparisons or next level discussion of why one therapy may be better suited under different conditions.

  • Rehabilitative therapies are the current standard of care after TBI. Information about dosing and what is standard would be a good addition to section 4 to contextualise the newer research.

  • A final suggestion I have is a change in title- some of the therapies are not focussed on replacement. There is a large amount of writing in this review covering IV MSC work. This title does not describe all research covered.

Other minor comments:

Please note an error in the affiliations with numbers not corresponding to author affiliations.

Author Response

This review covers a vast amount of emerging research to indicate progress in the area of stem cell therapies for TBI. Unfortunately, interpretation of the body of research presented here is not comprehensive or cohesive. Findings from the emerging research are not contextualised or compared. It is therefore difficult to derive any key take home messages or understanding of what research should be performed next, or even where the research landscape is currently at regarding both rehabilitation, cell therapies or combination interventions.

 Specific recommendations and reflections

Overall the tone of the paper infers that there is a lot of clinical work and that combined cell therapy for TBI is a well researched area. However this is a huge flaw of the review. There is a lack of robust clinical evidence and there remains vast cell combination candidates still being investigated preclinically. This isn’t to say that we aren’t on the cusp of existing clinical work to uncover the usefulness of these interventions. This is a promising area and research is extremely needed. But I think some of the language needs to be tempered throughout to discuss where the research is up to, and claims on safety and efficacy must be prefaced with the context the research has been carried out (e.g. preclinical).   

Response: We thank you for taking the time to review and critique this work. Having read the concerns and suggestions we agree that a certain amount of context is needed in order to illustrate the points we are trying to convey. To this end, we have rewritten and separated the scope and limitations within which this was presented in the introduction and the concluding remarks. Our intent was not to present this subject as a well-researched area either in the preclinical or clinical fields as we fully agree that there is currently much needed research in the fields presented here. Our intent was to convey that there have been many recent reviews in this field and we were not aiming to review/highlight the same research over again. Instead, we chose the most recent preclinical research in the field to highlight the advances that have been made and to present a combination of those as a hierarchy of treatments towards recovery following a TBI. We appreciate your comments and perspective and believe that the changes we have made will reflect that.

  • Line 23: Additional therapies such as pharmacological, stem cell, and rehabilitative have shown little to no improvement on their own, but when applied in combination have given encouraging results.

What about the STEMTRA trials? There have also been a number of clinical trials in this space using cord blood/bone marrow MSCs that have improved outcomes, even if effect sizes are small.

Response: We agree that current clinical trials show promise and that the recently published STEMTRA paper (Kawabori et. al., Neurology, 2021) provides further support for clinical translation. However, we feel that those very crucial clinical trials would rather deserve to be highlighted within their own context so as not to be confused within preclinical and basic research within the areas presented. Therefore, we intentionally left clinical trials out and have reflected that in the scope of this work. 

  • Line 56: Several very recent reviews document the use and efficacy of stem cell replacement therapies, pharmacological treatments, and combined stem cell and pharmacological therapies being tested and used

References here cover a number of neurological conditions. This section and the claims to follow are all based on documented work from animals. There needs to be a description of where the research is up to, or at least specify that the efficacy is being investigated remains in the preclinical stage and what has been done so far in the clinical area. For example, Bonsack (ref 7) is referred to on multiple occasions however the focus of the review discusses the important preclinical investigations that have primed some clinical research areas. The paper urges that more work is to be done, and clinical work to date is limited. Your next line of “That review concluded that the combination of pharmacotherapy and stem cell replacement therapies was able to provide the best outcomes compared to each of these on their own. Both of these therapies were effective together in reducing neuroinflammation while the stem cell therapy added the additional abilities of replacing lost brain cells, secreting neurotrophic factors, and recruiting cytokines and endogenous stem cells to the injury area. [7].” Will need to be significantly tempered to contextualise the findings.

Response: As stated in the response to the first comment, the objective of this manuscript was to not re-review what had already been published, but to present a new perspective on combinatorial treatments. We agree that the reviews covered additional conditions besides TBIs, yet that those paralleled the treatments highlighted for TBIs. For the sake of clarity and cohesion, the references have been edited and that line has been moved and revised to reflect a more contextualized statement.

The section referring to Bonsack has been revised to better explain the preclinical research and the conclusions that Bonsack derived from that work to place within the scope of this work.

  • Text under header 2 can be condensed. It is very repetitive from the introduction and i would recommend this be combined with the section prior to it. Instead this section likely needs an introduction to the candidate therapies themselves.

 Response: We completely agree and have changed this accordingly.

  • 1- Some statements presented here are false. “Whether this concern is valid only for intravenous application or for locally transplanted MSCs is not completely clear.”

It is well known than MSCs’ potential for tumour formation is based on a range of factors, not just route but also cell source, manufacture and manipulation rather than the route of administration. The referenced material is explicit about this e.g. Hasan 2017 discusses how tumor formation is a consideration, particularly for cells that have been cultured for long periods of time.

Response: This statement aimed to convey the range of possible factors and the need for further research. It has been removed and replaced with an expanded discussion/explanation of unintended consequences, a reference to ongoing clinical research, and the need for ongoing research.

  • My suggestion is to alter and reconsider some of your discussed rationale. The field has research on stem cell therapies from a range of sources, using a range of protocols which expand and characterise cells, mainly MSCs. This heterogeneity is a leading cause of the concerns, where you cannot standardise the risk or completely address safety concerns in the field without seeing a cell candidate through the entire pipeline in a streamlined and homogenous manner.

 Response: Agreed. We believe the changes made for this comment and the previous concern reflect this within the context of this review.

  • In section 2, you separate the use of cells transplanted via different routes contains no discussion of the reasoning behind the different delivery methods. Instead it is just a summary of the studies which use the methods. I would suggest discussion of the rationale for both uses of these methods, or why they may differ, particularly when discussing MSCs which as you say have a well defined secretome and known trophic effects.

This information is critical as you then lead into section 3, where it is important to understand if engraftment is even essential to see functional improvements and recovery or why this would be of benefit.

Response: As the objective of this paper is to convey the benefits of these combined therapies in some of the worst cases, namely stem cell therapies in initial TBI treatment, stem cells and their survival/integration as replacement therapies, and physical/cognitive therapies to bolster stem cell integration and synaptic connectivity. To this end, each of these research examples were incorporated. Providing three separate examples illustrates the different strategies currently being researched for TBI and moreover that a combination approach may address the shortcomings of each standalone approach for TBI intervention. Several changes were made to better convey this within the paper.  

  • By the end of section 3, it is unclear what method of cell delivery or use of scaffolds or combination therapy are preferred or show better outcomes. The sections do summarise the information but there are no comparisons or next level discussion of why one therapy may be better suited under different conditions.

 Response: Please see previous response.

  • Rehabilitative therapies are the current standard of care after TBI. Information about dosing and what is standard would be a good addition to section 4 to contextualise the newer research.

 Response: Changes have been made to better convey the point that preclinical research is lacking in rehabilitative therapies to support the integration of stem cell transplants and that the examples given illustrate the compensatory abilities that would better support stem cell transplants with the addition of rehabilitative therapies.  

  • A final suggestion I have is a change in title- some of the therapies are not focussed on replacement. There is a large amount of writing in this review covering IV MSC work. This title does not describe all research covered.

Response: The title was revised to: “Recent advances in stem cell therapies to address neuroinflammation, stem cell survival, and the need for rehabilitative therapies to treat traumatic brain injuries”

Other minor comments:

Please note an error in the affiliations with numbers not corresponding to author affiliations.

Response: Corrected. Thank you.

Reviewer 2 Report

The authors present a narrative review focusing on recent research and  advances in potential therapies for severe traumatic brain injuries (TBI), with an overview about stem cell therapies, advanced engineered biomaterials supporting stem cells transplantation, and also their potential role in rehabilitative therapies.

The topic is undoubtedly of great interest, considering the morbidity, mortality, and the socio-economical impact of TBI in general population. The review, despite in narrative form and not systematic, is timely and complete and potentially interesting for the readers, also considering the most updated literature currently available (see i.e. Lengel et al, Stem Cell Therapy for Pediatric Traumatic Brain Injury. Front Neurol. 2020 Dec 2;11:601286. doi: 10.3389/fneur.2020.601286, about the effects of stem cell treatments on histopathological and functional outcomes in models of pediatric brain injury).

I suggest to introduce a short paragraph about the methodological limitation related to the narrative review which, per se, could bring with them some bias.

Moreover, i suggest authors to consider a very recent paper (Caplan et al, Combination therapy with Treg and mesenchymal stromal cells enhances potency and attenuation of inflammation after traumatic brain injury compared to monotherapy. Stem Cells. 2020 Dec 23. doi: 10.1002/stem.3320) about the impact of immune suppressive regulatory T-cells in mesenchymal stromal cells infusion for severe TBI.

Author Response

The authors present a narrative review focusing on recent research and  advances in potential therapies for severe traumatic brain injuries (TBI), with an overview about stem cell therapies, advanced engineered biomaterials supporting stem cells transplantation, and also their potential role in rehabilitative therapies.

The topic is undoubtedly of great interest, considering the morbidity, mortality, and the socio-economical impact of TBI in general population. The review, despite in narrative form and not systematic, is timely and complete and potentially interesting for the readers, also considering the most updated literature currently available (see i.e. Lengel et al, Stem Cell Therapy for Pediatric Traumatic Brain Injury. Front Neurol. 2020 Dec 2;11:601286. doi: 10.3389/fneur.2020.601286, about the effects of stem cell treatments on histopathological and functional outcomes in models of pediatric brain injury).

I suggest to introduce a short paragraph about the methodological limitation related to the narrative review which, per se, could bring with them some bias.

Response: Thank you for taking the time to review our article. We have made several changes to the article to more focus and contextualize what has been written as suggested. Several other revisions have been made throughout the article to further focus on the themed therapies of the paper.

Moreover, i suggest authors to consider a very recent paper (Caplan et al, Combination therapy with Treg and mesenchymal stromal cells enhances potency and attenuation of inflammation after traumatic brain injury compared to monotherapy. Stem Cells. 2020 Dec 23. doi: 10.1002/stem.3320) about the impact of immune suppressive regulatory T-cells in mesenchymal stromal cells infusion for severe TBI.

Response: We thank you for the suggestion and are currently looking at that paper in particular the timing and multi-dosing aspects. However, this article does not fit within the scope of our current review.
